# Low-Temperature Transient Liquid Phase Bonding Technology via Cu Porous-Sn58Bi Solid–Liquid System under Formic Acid Atmosphere

**DOI:** 10.3390/ma16062389

**Published:** 2023-03-16

**Authors:** Siliang He, Bifu Xiong, Fangyi Xu, Biyang Chen, Yinhua Cui, Chuan Hu, Gao Yue, Yu-An Shen

**Affiliations:** 1Guangxi Education Department Key Laboratory of Microelectronic Packaging & Assembly Technology, School of Mechanical & Electrical Engineering, Guilin University of Electronic Technology, Guilin 541004, China; siliang_he@guet.edu.cn (S.H.);; 2Institute of Semiconductors, Guangdong Academy of Sciences, Guangzhou 510650, China; 3Guilin Fuda Co., Ltd., Guilin 541199, China; 4Department of Materials Science and Engineering, Feng Chia University, Taichung 407, Taiwan

**Keywords:** TLP bonding, porous Cu, formic acid, soldering, intermetallic compounds

## Abstract

This study proposes a low-temperature transient liquid phase bonding (TLPB) method using Sn58Bi/porous Cu/Sn58Bi to enable efficient power-device packaging at high temperatures. The bonding mechanism is attributed to the rapid reaction between porous Cu and Sn58Bi solder, leading to the formation of intermetallic compounds with high melting point at low temperatures. The present paper investigates the effects of bonding atmosphere, bonding time, and external pressure on the shear strength of metal joints. Under formic acid (FA) atmosphere, Cu_6_Sn_5_ forms at the porous Cu foil/Sn58Bi interface, and some of it transforms into Cu_3_Sn. External pressure significantly reduces the micropores and thickness of the joint interconnection layer, resulting in a ductile fracture failure mode. The metal joint obtained under a pressure of 10 MPa at 250 °C for 5 min exhibits outstanding bonding mechanical performance with a shear strength of 62.2 MPa.

## 1. Introduction

Third-generation/wide bandgap semiconductors are the materials used in electronic devices that have a larger bandgap than traditional silicon-based semiconductors. These materials include compounds, such as gallium nitride (GaN) and silicon carbide (SiC), and they possess a large bandgap available for the critical operation conductions at high breakdown voltages and high current density. Additionally, those compound semiconductors with good thermal stability and thermal conductivity can resist a higher working temperature than the traditional semiconductor (Si) [1,2,3]. Therefore, those third-generation semiconductors with continuous development for various applications (electric vehicles, power control, the fifth-generation mobile network, and communication satellites) [4,5,6,7,8]. However, when their working temperature can achieve up to 200~450 °C, the thermal stability and high-temperature resistance of the interconnecting materials must also be upgrading and developing, especially to replace the traditional solder interconnections not available for working at a temperature higher than 200 °C [9,10].

Interconnections using Cu/Ag particle sintering have been developed in some studies. However, such a method generally requires bonding temperatures above 300 °C and bonding times greater than 30 min [11,12,13]. As the world moves towards reducing energy consumption and achieving carbon neutrality, there is a significant need for developing reliable joints of electronic packaging that can be fabricated at low temperatures and withstand high temperatures [14,15]. Recent advancements in low-temperature transient liquid phase bonding (TLPB) have resulted in the development of die package joints capable of withstanding higher service temperatures at lower bonding temperatures [16,17]. The basic principle of low-temperature TLPB is that the intermediate layer bond quickly melts into a liquid phase, and then diffuses and reacts with the metal to be bonded in a solid–liquid phase to form a corresponding intermetallic compound (IMC) [18,19,20,21,22]. The mixture of low-melting alloys (Sn-based solders) and high-melting-point metal Cu metallurgical system is widely used in the TLPB process because of the activated interfacial reaction. After TLPB, the joints consisting of high-melting IMC and Cu can be applied for high-operating-temperature devices. However, how to keep lowering the bonding temperature and time, and simultaneously improve the bonding strength, is a concern.

The eutectic Sn-58Bi alloy performs good wettability with bonding metallic substrates [23,24,25,26]. The eutectic Sn-58Bi alloy contains precipitated Bi that creates numerous phase boundaries and serves as a secondary phase within the Sn matrix [27,28,29]. This effectively strengthens the Sn matrix and enhances its mechanical properties. Furthermore, Sn-58Bi alloy possesses a low melting temperature and great mechanical strength [30,31,32]. Therefore, Sn-58Bi alloy is a good candidate for serving as one of the materials for low-temperature TLPB.

Porous Cu foil refers to a type of copper foil that has a three-dimensional porous structure and has a wide range of applications in various fields, such as in energy storage devices (e.g., batteries, supercapacitors) and catalysts [33,34]. Porous Cu foil can also be used in TLPB as an interlayer material. When porous Cu foil is used as an interlayer material in TLPB, the Cu foil’s porous structure facilitates the wetting and spreading of the liquid phase, promoting the formation of a uniform bond. Moreover, the high surface area of the porous Cu structure enables efficient diffusion of the liquid phase, thereby accelerating the formation of the IMC. These desirable properties make porous Cu foil an excellent candidate for enhancing bonding time and reliability.

During the preparation and packaging process, oxide films tend to form on the surface of Cu substrates and Sn58Bi solder, impeding atomic diffusion and weakening bonding efficiency. This problem is especially pronounced in porous Cu foil with high chemical activation and surface area. Flux is typically used to remove these oxide films during the bonding process. However, due to the porous structure of Cu foil, especially one with high porosity and large specific surface area, it can be challenging for the flux in solder paste to fully infiltrate and remove the oxide films. To address this challenge, the use of a formic acid (FA) atmosphere has been investigated to facilitate reduction–oxidation reactions and improve the soldering reaction between solders and Cu substrates. The FA atmosphere can effectively remove oxide films in porous Cu while also protecting bonding materials from oxidation at high bonding temperatures. Overall, the use of a FA atmosphere offers a promising approach for enhancing bonding efficiency and reliability in porous Cu foil applications [35,36,37].

Although the use of Sn-58Bi solder, porous Cu, and FA bonding/soldering has been studied for electronic interconnects, there is a scarcity of research on the low-temperature transient liquid phase (TLP) reaction and TLPB reliability when utilizing both Sn-58Bi solder and porous Cu foil under a FA atmosphere.

In this study, TLPB of Cu/Sn58Bi-porous Cu-Sn58Bi/Cu was assembled at 250 °C under a FA atmosphere by different bonding times and pressures. The bonding strength and deformation behavior in the TLPB joints were studied. Additionally, the capillary phenomenon between the Sn-58Bi solder and the porous Cu skeleton was explored.

## 2. Experimental Procedures

### 2.1. Materials

In this study, the Cu/Sn58Bi-porous Cu-Sn58Bi/Cu structure was utilized to establish the intermetallic compound (IMC) bonds. The bottom and top bonding substrates were made of high-purity Cu plates (99.9%) measuring 12 mm × 20 mm × 2 mm and 4 mm × 4 mm × 2 mm, respectively. A micro-sized porous Cu foil with a thickness of 1 mm and 96.43% porosity was used as the bonding interlayer between Sn58Bi solders (Senju Co., Ltd., Huizhou, China).

### 2.2. Experiment and Analysis

Figure 1a shows the schematic of the TLP bonding process using the Sn58Bi solder/porous Cu foil/Sn58Bi solder sandwich structure under a FA atmosphere. The Cu plates and porous Cu foil were immersed in 8% HCl solution for 5 and 3 min to deduct the surficial oxidation layer, respectively. Then, they were cleaned by ethanol ultrasonic pool for 5 min to remove the surficial organic matter. A 0.2 mm thick Sn58Bi solder paste was brushed on the top and bottom surfaces of the porous Cu foil, and then the Sn58Bi/porous Cu/Sn58Bi sandwich structure formed. Before the bonding process, the pre-treated sample was placed on the platform in a sintering machine and the gaseous FA was introduced into the chamber, as shown in Figure 1b,c.

The sample prepared at 250 °C in air atmosphere for 5 min was selected as the reference in order to illustrate the effect of FA atmosphere. In addition, the effects of other process parameters upon FA, such as bonding time, bonding temperature, and applied external pressure on the shear strength of metal welded joints, were also studied. Table 1 shows the specific experimental parameters of each sample, and the heating rate was set as 2 °C/s constantly. Shear test was performed after the bonding process in order to evaluate the shear strength of the joint using a shear tester (MFM1200), and the testing speed was 200 μm/s. Figure 2 shows the shear strength testing schematic diagram. The calculation method for shear strength is shown in Formula (1), and the average value of three joints is taken as the measured shear strength value. The cross-sectional morphologies and structures of the joints under different process parameters were studied by optical microscope (OM) and scanning electron microscope (SEM, Quanta FEI 450), with the accelerating voltage of 5 kV. The failure modes of the joints during shear strength test were also discussed.
(1)$$\tau =\frac{F}{S}$$

Formula:-*F*—Shear load. The unit is N;-*S*—Metal joint contact area. The unit is m^2^;-*τ*—Shear strength. The unit is MPa.

**Table 1 materials-16-02389-t001:** Summary of heating processes.

Process Number	Temperature (°C)	Bonding Time (min)	Atmosphere	Pressure (MPa)	Catalytic
1	250	5	Air	No	-
2	250	5	FA	No	-
3	250	10	FA	No	-
4	250	20	FA	No	-
5	250	5	FA	3	-
6	250	5	FA	5	-
7	250	5	FA	10	-
8	250	5	FA	20	-

**Figure 2 materials-16-02389-f002:**
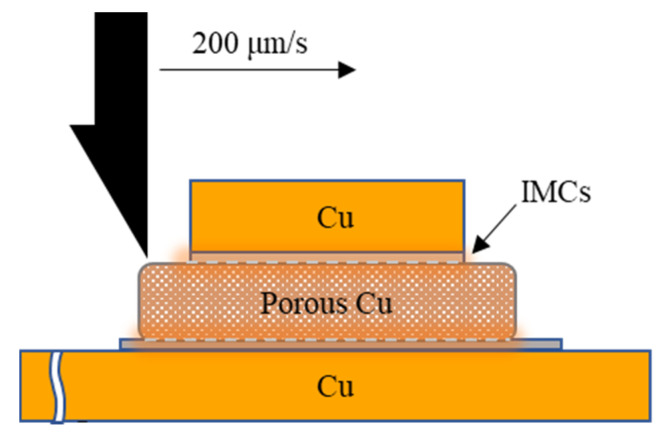
Schematic diagram of shear strength test.

## 3. Results and Discussion

### 3.1. Effect of Bonding Atmosphere on the Shear Strength of Joints

Figure 3a shows the shear strength of TLPB joints under the FA and air atmosphere. The shear strength of the latter (12.9 MPa) is lower than the former (15.8 MPa), and their fracture morphologies are shown in Figure 3b,c. Figure 3b exhibits that a small amount of solder adheres to the Cu substrate after the FA TLPB. That indicates the formation of an intermediate solder layer providing a good bonding to the Cu substrate. Conversely, the overall peeling fracture was observed on the fracture surface of the TLPB joint under the air atmosphere, exhibiting the mechanical connection between the porous Cu foil, Sn58Bi, and substrate. The FA removed the oxide film on the surface of the bonding materials, activating the TLPB. The Sn formate can be formed on the diffusion interfaces between the Sn-58Bi and Cu substrates/porous Cu during the chemical reaction of the FA the Sn oxide at 150 °C (Formulas (2)–(4)):SnO + 2HCOOH(g) → Sn(HCOO)_2_ + H_2_O(g) (2)SnO_2_ + 2HCOOH(g) → Sn(HCOO)_2_ + H_2_(g) + CO_2_(g) (3)Sn(HCOO)_2_ → Sn + H_2_(g) + CO_2_(g) (4)

Therefore, the oxide-free interface activated the soldering and bonding process [38,39,40]. A similar reduction reaction also occurred between the FA molecule and the CuO, and the Cu substrate and porous structure could be protected from oxidation. Conversely, the air bonding can lead serious oxidation to reduce the bonding strength.

### 3.2. Effect of Bonding Time on the Shear Strength of Joints

During the low-temperature TLPB process, there are four stages including solder melting, solid–liquid phase interaction, isothermal solidification, and homogenization of the solder joint. They require enough time for the diffusion processes. If the bonding time is not long enough, the incomplete TLPB causes a low bonding strength. On contrary, an excessively long bonding time induced the structure coarsening of the materials and the overgrowth of the brittle IMC, which dramatically reduced the bonding strength of the joints. Therefore, the TLPB time was optimized by examining the shear strength of the joints assembled by TLPB under the FA atmosphere at 250 °C for 5, 10, and 20 min.

Figure 4 summarizes the shear strengths of the joints. The shear strength was increased with the increase in the TLPB time, and the highest shear strength (21.4 MPa) was achieved by the bonding time of 20 min, with an enhancement of 38.96% compared with that of 5 min. However, the shear strength of the solder joints increases slowly with the bonding time when it is more than 10 min.

In the reference, the lead-free solder joint by soldering possessed a shear strength of ~15 MPa [41,42]. In our results, the strength by the bonding time of 5 min can be regarded as well completed, and the increases in the bonding strengths with the bonding times can enhance the TLPB reliability. Therefore, the combination of the Sn58Bi, porous Cu film, and FA atmosphere is successful. Additionally, it is worth noting that the bonding strength of our joints may be a little lower than that of Cu/Sn58Bi/Cu solder joints at room temperature; it still shows the advantage of much shorter soldering time while comparable bonding strength at elevated temperatures.

Figure 5 shows the SEM images of fracture surface on the bottom Cu substrate side under different heating times. A few filled pores of porous Cu foil and the obvious skeleton of the porous Cu were observed after 5 min of TLPB. Figure 5a,b show a magnified image of the filled pores, in Figure 5a, and the porous structure can be observed, indicating the incomplete reaction of the low-temperature TLPB. Conversely, the porous structure of the Cu foil was filled by Cu-Sn IMCs in the 20 min TLPB case, as shown in Figure 5e,f. The pores in the porous Cu are gradually filled with the increase in the bonding time. The Cu_3_Sn (74.05 at % Cu, 25.95 at % Sn, by EDS analysis) preferred to generate on the copper foam skeleton and the Cu_6_Sn_5_ (57.03 at % Cu and 42.97% at % Sn, by EDS analysis) formed at the edge of the porous Cu skeleton, indicating the diffusion-controlled reaction between Cu and Sn during the soldering process. Remarkably, the structures were observed in both joints, as shown in Figure 5c,g. In addition, Figure 5d,h show that the amount of the generated IMCs in the joint of 20 min TLPB is more than that of 5 min TLPB. That was believed to enhance the bonding strength of the joint [43,44,45]. Addition of porous Cu, it seems, cannot decrease the growth rate of IMCs, but it can suppress the formation of voids at the soldering interface and improves the shear strength of the solder joints during the aging process, according to Liu’s report [46].

### 3.3. Effect of Applied Pressure on Shear Strength of Joints

During the liquid bonding process, defects such as pores and microcracks are formed at the bonding interface generally due to liquid phase volatilization, solid–liquid phase transition, and interfacial reaction. These defects might be the original source of fatigue cracks, change the propagation mode and rate of the cracks, and then accelerate the expansion of the microcracks under shear stress, decreasing shear strength and the service reliability of the solder joint significantly. The applying of local pressure onto the chip or substrate during TLP bonding can reduce the porosity of the bonding interface significantly, which in turn improves the bond strength of the solder joints [47,48]. Herein, the effect of the bonding pressure of the low-temperature TLPB under the FA atmosphere was also investigated. The bonding process was carried out at 250 °C for 5 min with the applied pressure of 0, 3, 5, 10, and 20 MPa. Figure 6 shows the shear strength of the TLPB joints under various pressures. The bonding time played the key role in the TLPB strength. The shear strengths of the joints were 15.4, 34.2, 48.4, 62.2, and 71.2 MPa via the bonding pressure of 0, 3, 5, 10, and 20 MPa, respectively. Enhancement of the bonding strength was 362% from the bonding pressure of 0 MPa to that of 20 MPa.

According to the reference, a bonding strength of approximately 15–25 MPa was achieved in the Cu/Sn/Cu joint under a bonding pressure of 5 MPa [49]. This strength is comparable to that observed at 3 MPa in Figure 6. Moreover, it should be noted that the TLPB joint demonstrated superior high-temperature performance compared with the Cu/Sn58Bi/Cu solder joint. Therefore, the findings of this study suggest the promising potential of TLPB utilizing Sn58Bi solder, porous Cu, and FA.

According to the morphologies of the fractured joints after shear test shown in Figure 7, the micropores of the fractured porous Cu skeleton in the joints with bonding pressures is significantly reduced compared with that with bonding pressure (Figure 5a). The pores in the porous Cu are progressively filled. Additionally, the micropores pores left reduced with the increase in the bonding pressure, and the edge of the joint by 5 MPa bonding was integrated with the substrate, and only a small amount of foam-like structure remained in the middle of the joint. Fractures of the joints with 3 and 5 MPa bonding were brittle fractures, and that with 20 MPa bonding possessed a ductile–brittle fracture due to the observation of both tearing edge and smooth cuts. This also indicates the latter possesses the highest bonding strength.

Figure 8 shows the EDS images of the fractured joints after shear test formed by different bonding pressures. The voids in the bonding area gradually decreased with the increase in the bonding pressure. In the joints by the 3 and 5 MPa bonding, the Sn and Bi obviously have a stronger and more uniform distribution in contrast to Cu, indicating the initial formation of the metallurgical bonding at the interface (Figure 8a,b). As the pressure increased to 20 MPa, the solder could react with the porous Cu and the Cu substrates thoroughly; consequently, the pores in the joint almost disappeared after the strong metallurgical bonding at the interface (Figure 8c). The application of bonding pressure can reduce the soldering porosity significantly through this low-temperature TLPB, improve the contact areas among the solder, porous Cu foil, and the substrate, and then promote the interdiffusion reaction at the interface. Remarkably, the pressure should be controlled in an appropriate range with the aim to improve the shear strength significantly while avoiding damage to the chips.

## 4. Conclusions

In this study, a highly effective TLPB process utilizing a sandwich structure of Sn58Bi solder/porous Cu foil/Sn58Bi solder under FA atmosphere was developed. The rapid reaction between Sn58Bi and Cu within the sandwich structure, followed by the instant formation of intermetallic compounds such as Cu_6_Sn_5_ and Cu_3_Sn, resulted in the establishment of a strong metallurgical bond at the interface. The effects and mechanisms of various parameters, such as bonding atmosphere, time, and external pressure on the shear strength of the metal joints were thoroughly investigated. The results are summarized below:(a)The FA could reduce the oxides of solder and Cu; thus, the interconnection between porous Cu and Cu substrate was fulfilled, and the shear strength was approximately 22.48% higher than that obtained under the air atmosphere.(b)The strength of metal joints increased with increases of bonding time significantly within 10 min due to the rapid infiltration and reaction between Sn58Bi solder and porous Cu layer with the help of FA. Beyond that, the shear strength increased slowly with the increase in bonding time, which might be the full and complete reaction between them within 10 min.(c)The applied pressure showed obvious influence on the shear strength of the joints, and it could be promoted to 34.2, 48.4, 62.2, and 71.2 MPa with the application pressure of 3, 5, 10, and 20 MPa, respectively, while the shear strength was only 15.4 MPa in the non-pressure case. The mechanism should be ascribed to the accelerated infiltration and reaction at the Sn58Bi solder/porous Cu interface and the compact structure at the interfacial zone upon pressure.

These findings prove that the LTPB system consisting of the Sn58Bi, porous Cu film, and FA is promising to realize low-temperature bonding with high-temperature applications.

## Figures and Tables

**Figure 1 materials-16-02389-f001:**
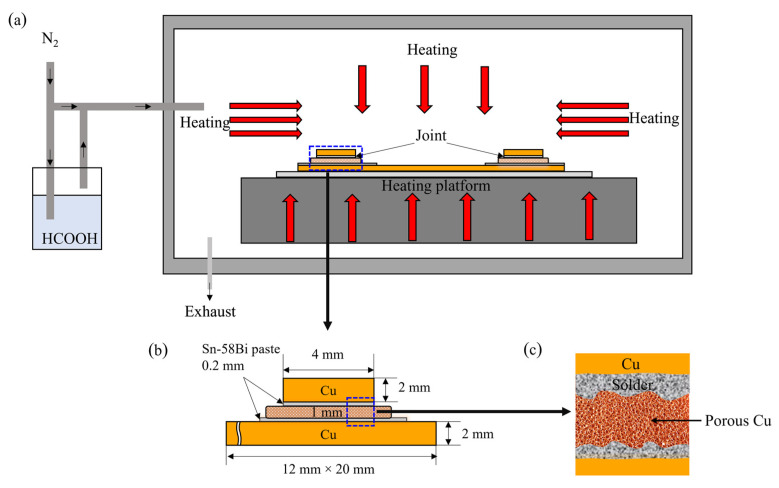
Schematic graph of the method in this study: (**a**) bonding process under FA atmosphere; (**b**) Sn58Bi solder/micro-size porous Cu foil/Sn58Bi solder sandwich structure; (**c**) partial enlarged view of the sandwich structure.

**Figure 3 materials-16-02389-f003:**
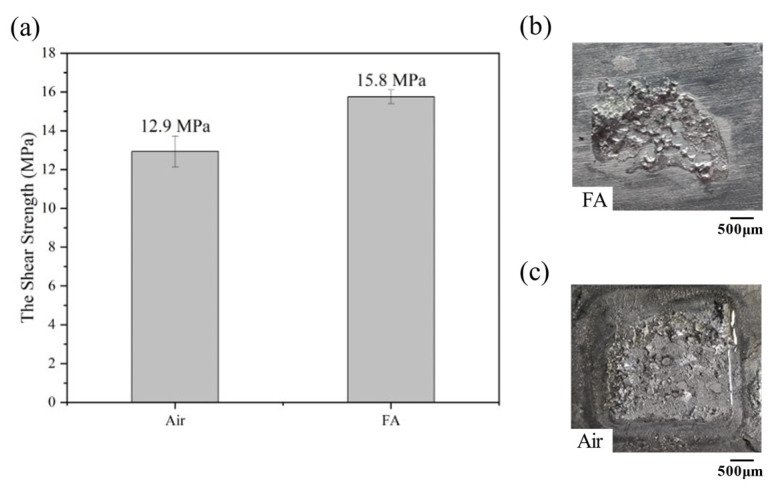
The shear strength and the OM morphologies of the joint under different welding atmospheres: (**a**) statistical data of shear strength; (**b**) the failed solder joint in FA atmosphere after shear strength test; (**c**) the failed solder joint in air after shear strength test.

**Figure 4 materials-16-02389-f004:**
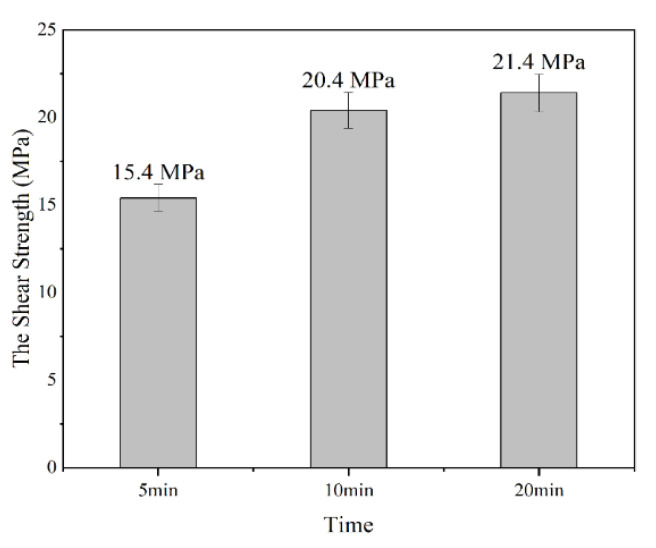
The shear strength at different times.

**Figure 5 materials-16-02389-f005:**
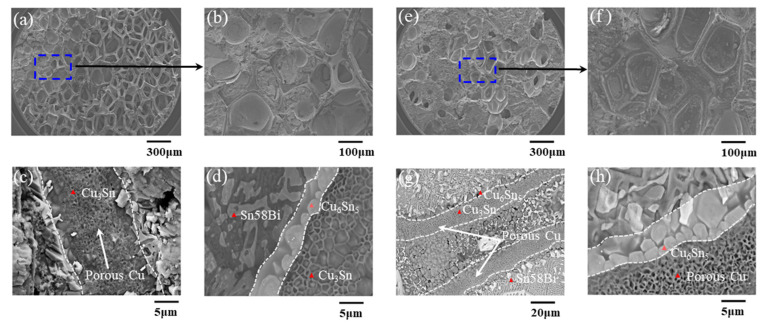
SEM images of joints bonded for different times: (**a**–**d**) 5 min and (**e**–**h**) 20 min.

**Figure 6 materials-16-02389-f006:**
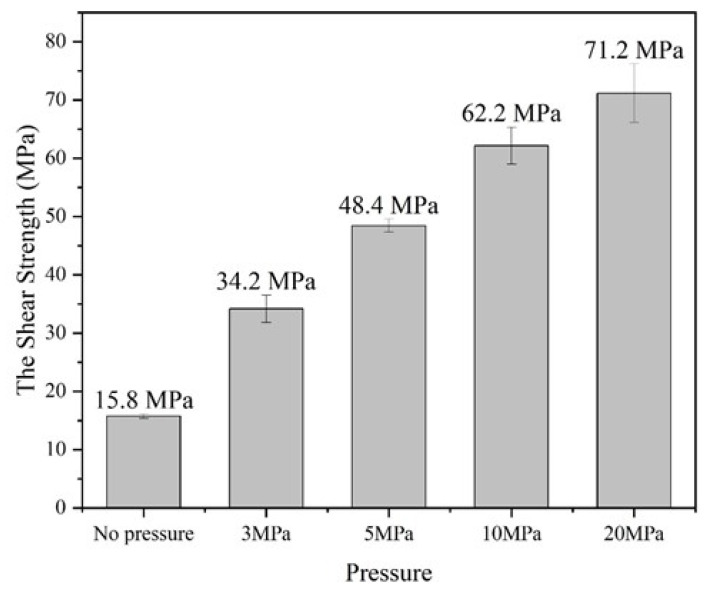
Shear strength of the joints obtained under different pressures.

**Figure 7 materials-16-02389-f007:**
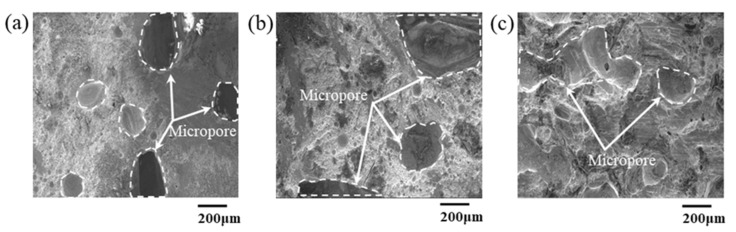
SEM micrographs of the fractured joints obtained under different pressures after shear failure test: (**a**) SEM images at 3 MPa; (**b**) 5 MPa; (**c**) 20 MPa.

**Figure 8 materials-16-02389-f008:**
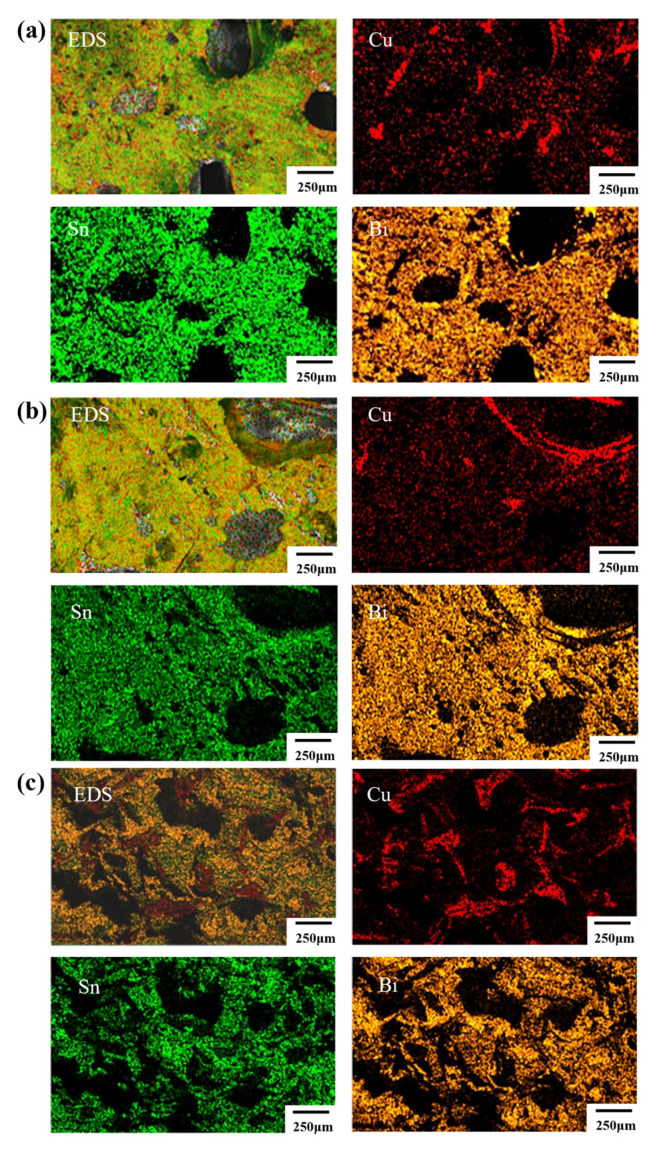
The EDS mapping results of the fractured joints obtained under different pressures after shear failure test: (**a**) EDS picture under 3 MPa pressure; (**b**) EDS picture under 5 MPa pressure; (**c**) EDS picture under 20 MPa pressure.

## Data Availability

Unavailable due to privacy.

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
