# Peer review of "Low-Temperature Transient Liquid Phase Bonding Technology via Cu Porous-Sn58Bi Solid–Liquid System under Formic Acid Atmosphere"

_materials, 2023, doi:10.3390/ma16062389_

Round 1

Reviewer 1 Report

The paper ”Low-Temperature Transient Liquid Phase Bonding Technology via Cu Porous-Sn58Bi Solid-liquid System under Formic Acid Atmosphere” is well written and is suitable for publication in Materials journal after some minor corrections.

1. The introduction should contain the influence of Cu, Sn and Bi in terms of correlation between microstructure, mechanical properties, and corrosion resistance.

2. Also, please compare with other types of bonding or coating, like APS. Recommended article: 10.3390/coatings10121186.

3. Add SEM parameters.
 4.figure 5: Improve the contrast of the images.

 5. please summerize the conclusions.

 Rest is ok

Reviewer 2 Report

Good work done by the author however improvement must be done before publication.

I suggest a major revision. Following are the suggestions for improvement.

1. English should be improved
2. introduction needs to be improved. The author should clarify the novelty using the existing studies. Several studies has already worked on this topic. The justification should also be given. 

3. Author also overlooked the existing closely related study on Sn-58Bi in 2022 and 2023 ie.

https://link.springer.com/article/10.1007/s10854-022-09465-2

https://link.springer.com/article/10.1007/s10854-022-09035-6

https://link.springer.com/article/10.1007/s10854-022-09028-5

4. The Cu plates 84 and porous Cu foil were immersed in 8 % HCl solution for 5 mins and 3 mins to deduct 85 the surficial oxidation, respectively.

why specifically for 5 and 3 min

5. The sample prepared at 250 °C 

what is the reason using 250 please explain using the above ref. 

6. Figure 4 summarizes the shear strengths of the joints. 

this is just the summary of the author's findings and there is no discussion comparing with past studies. The author should compare the strength of the samples with past studies in a table. The researcher has already done work on shear strength, tensile strength, ball shear, etc using SnBi and different compositions of the solder joint. A separate table can be prepared in which different values of the past can be compared with current shear strength values.

7. Fig 4-7 need discussion and comparison with the existing study. 

Reviewer 3 Report

The manuscript presents a research on effect of atmosphere, bonding time and applied external pressure on the microstructure and shear strength of Cu/Sn58Bi/porous-Cu/Sn58Bi/Cu joints. This is an interesting topic but the manuscript needs to be much improved to be accepted for publication.

First of all, this is not the first time that someone studied mechanical properties of Cu/Sn58Bi/Cu joints reinforced with porous Cu. Liu et al. in ” Improvement on the mechanical properties of eutectic Sn58Bi alloy with porous Cu addition during isothermal aging” DOI 10.1088/2053-1591/ac10d5 studied the effect of porous copper particles on properties of joints. Simply having in mind potential practical application, it seems their approach i.e. mixing Sn58Bi solder paste with porous Cu particles is smarter. They avoid the problem of solder paste penetration through the pores and it is probably easier to apply such solder. Please refer to the above paper in your manuscript.

Further questions and comments, merely intended to help the Authors improve their manuscript:

1.       In the lines 71-74, and only in these lines, LTLPB acronym is used, which is not defined earlier in the text. Because it is not used elsewhere in the text simply write “low-temperature TLPB” here.

2.       Line 79 states that “96.43% Porosity of the micro-size porous Cu foil” was used. Are you sure that the porosity was exactly 96.43%? If the porosity is so high then this material is a foam rather than a foil. What is the average size of the pores?

3.       Sn58Bi is near-eutectic alloy with melting point of 139oC, then why the soldering is carried out at 250oC? With such a high porosity of Cu foam the liquid solder should penetrate the foam easily a lower temperatures.

4.       In the lines 142-144 it is stated that there are four stages of the low-temperature TLPB process: solder melting, solid-liquid phase interaction, isothermal solidification, and homogenization of the solder joint. The heating rate is 2oC/min (line 98), which means that molten solder is in contact with Cu already 55 min before 250oC is reached. It is safe to assume that solder interacted with Cu during heating and there was no liquid left (i.e. the joint solidified) before 250 oC was reached. Please give your comment on this.

5.       Please, expand the discussion in sections 3.1-3.3. For example the mechanical properties could be discussed with results of others (other authors), who studied Cu/Sn58Bi/Cu joints, where solder was “pure” Sn58Bi i.e. without reinforcements.

6.       The subsection 3.4, i.e. the effect of Pt-catalyst, seems unnecessary and its results not too well explained. Maybe save it for another paper.

7.       Please add a scale bar to micrographs in Fig. 3(b) and 3(c).

8.       Fig. 5: Please explain what plane do these micrographs represent i.e. are they parallel to the cross-section of the interface (such as Fig. 1c) or parallel to Cu plates?

9.       In Fig. 5 there are some red points labeled Cu6Sn5, Cu3Sn and so on. How did you identify these phases? Did you just do the EDS analysis at these points?

10.   Micrographs in Fig. 8 (in the PDF for review) are particularly blurry and difficult to read – please check resolution of the uploaded image. Micrographs in Fig. 5 are only little better.

11.   The manuscript needs to be carefully proofread by a language professional or a native English speaking colleague.

Round 2

Reviewer 3 Report

The manuscript has improved and the Authors took into consideration most of comments from the previous review. The manuscript could be accepted for publication in its present form. However, I still think that this manuscript would benefit if in the discussion part the Authors referred to Liu et al., ”Improvement on the mechanical properties of eutectic Sn58Bi alloy with porous Cu addition during isothermal aging” Mater. Res. Express 8 076302, DOI 10.1088/2053-1591/ac10d5.
